# Memory Disagreement: A Pseudo-Labeling Measure from Training Dynamics for Semi-supervised Graph Learning

## ABSTRACT

In the realm of semi-supervised graph learning, pseudo-labeling is a pivotal strategy to utilize both labeled and unlabeled nodes for model training. Currently, confidence score is the most frequently used pseudo-labeling measure, however, it suffers from poor calibration and issues in out-of-distribution data. In this paper, we propose memory disagreement (MoDis for short), a novel uncertainty measure for pseudo-labeling. We uncover that training dynamics offer significant insights into prediction uncertainty —*if a graph model makes consistent predictions for an unlabeled node throughout training, the corresponding predicted label is likely to be correct. Thus, the node should be suitable for pseudo-labeling.* We implement MoDis as the entropy of an accumulated distribution that summarizes the disagreement of the model's predictions throughout training. We further enhance and analyze MoDis in case studies, which show nodes with low MoDis are suitable for pseudo-labeling as these nodes tend to be distant from boundaries in both graph and representation space. We design MoDis based pseudo-label selection algorithm and corresponding pseudo-labeling algorithm, which are applicable to various graph neural networks. We empirically validate MoDis on eight benchmark graph datasets. The experimental results show that pseudo labels given by MoDis have better quality in correctness and information gain, and the algorithm benefits various graph neural networks, achieving an average improvement of 3.11% and reaching up to 30.24% when compared to the wildly-used uncertainty measure, confidence score. Moreover, we demonstrate the efficacy of MoDis on out-of-distribution nodes. All code will be released after reviewing, according to the conference policy.

## CCS CONCEPTS

• **Computing methodologies → Learning latent representations**; **Neural networks**; • **Theory of computation → Social networks**; **Semi-supervised learning**.

## KEYWORDS

Graph Neural Networks, Epistemic Uncertainty, Self-Training

**ACM Reference Format:**
Anonymous Author(s). 2018. Memory Disagreement: A Pseudo-Labeling Measure from Training Dynamics for Semi-supervised Graph Learning. In *Proceedings of Make sure to enter the correct conference title from your rights confirmation emai (Conference acronym 'XX).* ACM, New York, NY, USA, 14 pages. https://doi.org/XXXXXXX.XXXXXXX

## 1 INTRODUCTION

Pseudo-labeling is a widely adopted strategy in semi-supervised graph learning (SSGL) to overcome the challenge of very limited number of labeled nodes in practice [9, 17, 21, 35]. It uses predicted labels as pseudo-labels to augment the limited labeled nodes, thereby decreasing epistemic uncertainty and facilitating graph model training. Theoretically, pseudo-labeling is a kind of entropy minimization that seeks to reduce the density of data embeddings near decision boundaries, thereby promoting the establishment of robust boundaries in low-density embedding regions [6, 13].

The effectiveness of pseudo-labeling based SSGL heavily relies on the correctness of pseudo-labels. Incorrect pseudo-labels can significantly degrade model performance because they inject noise and misleading patterns into graph model. Existing methods [34, 42–44] usually employ uncertainty measures of prediction, to act as proxies for generalization risk, in selecting pseudo-labels, such as confidence score and Bayesian uncertainty. However, these uncertainty measures are not effective indicators of the generalization risk. This issue has been highlighted in recent studies [27, 40].

In literature, the most frequently used uncertainty measure for selecting pseudo-labeled nodes is *confidence score* [5, 15, 35]. However this measure suffers from poor calibration, whereby high confidence scores are often assigned to incorrect predictions, resulting in incorrect pseudo-labels [14]. Here, the calibration measures the discrepancy between the model's confidence score in its predictions and the actual correctness of these predictions [8]. Despite several attempts to calibrate models for SSGL, the problem remains largely unresolved [38, 39]. Additionally, there is an argument that the confidence score should not be trusted for out-of-distribution data [10]. The confidence score is susceptible to manipulation by adversarial examples [24], and it is even possible to produce an incorrect prediction with arbitrarily high confidence by magnifying the input to a ReLU network [16]. The weaknesses further reduce the reliability of confidence score for pseudo-labeling. *Therefore, it is of great value to explore new uncertainty measures, so as to provide robust alternatives for pseudo-labeling in SSGL.*

In this paper, we propose a novel uncertainty measure, named **Memory Disagreement** (MoDis for short), which aims to identify pseudo-labeled nodes for semi-supervised graph learning. The MoDis is defined on training dynamics, which captures the disagreement among model predictions at different training epochs. The rationale behind using MoDis as a measure of prediction uncertainty is intuitive—*if a model makes consistent predictions for an unlabeled node throughout training, the corresponding predicted label is likely to be correct. Thus, the node should be a suitable candidate for pseudo-labeling. In contrast, if predictions fluctuate significantly, there is a high generalization risk associated with the predictions.*

Our basic idea is inspired by two intriguing observations about training dynamics. One observation is the complexity of decision boundary of deep networks gradually increases with the number

of training epochs, which suggests the networks first learn simple hypotheses and gradually learn complex hypotheses [2]; Another observation is deep networks first learn simple and general patterns in typical samples, during training, before fitting noise [31]. *From the two observations, we can deduce that typical samples, characterized by simple and general patterns, are those predicted consistently by model during training.* These typical samples are suitable for pseudo-labeling due to two properties: (i) **High correctness.** Typical samples characterized by simple patterns are likely to be predicted correctly. (ii) **Significant information gain.** Typical samples with general pattern are informative and can benefit to establishing decision boundaries for other samples. We validated this hypothesis in our first experiment on graph datasets.

We implement the memory disagreement MoDis by modeling the prediction uncertainty as the entropy of an accumulated prediction distribution that summarizes the disagreement of model's predictions throughout training. We further improve the discriminative ability of MoDis by incorporating the trajectory of softmax distribution of predictions during training. The softmax distribution can be considered as data uncertainty, although it is not a reliable estimator of prediction uncertainty [12]. Our case studies demonstrate that nodes with low MoDis tend to be distant from boundaries in both graph and representation space. The characteristics indicate the nodes are suitable for pseudo-labeling in SSGL.

We then design MoDis based pseudo-label selection algorithm and the corresponding pseudo-labeling algorithm, both of which are applicable to various graph neural networks. Our validations are conducted on 8 benchmark graph datasets, which show: (i) Pseudo labels given by MoDis have high quality in correctness and information gain; (ii) Graph neural networks equipped with MoDis based pseudo-labeling consistently yield superior performance, achieving an average improvement of 3.11% and reaching up to 30.24% compared to the wildly-used confidence score. (iii) MoDis is effective for out-of-distribution nodes.

In summary, our contribution in this paper is three-fold:

- We uncover that training dynamics provide significant insights into prediction uncertainty, from which we propose a novel uncertainty measure, memory disagreement MoDis, to identify pseudo-labeled nodes in SSGL.
- We analyze the rationale of MoDis in case studies, and accordingly design MoDis based pseudo-labeling algorithm that are applicable to various graph neural networks.
- We conduct extensive experiments to thoroughly validate the MoDis on eight benchmark graph datasets, establishing a new state-of-the-art for pseudo-labeling in SSGL.

## 2 PROBLEM DEFINITION

We first present a definition of SSGL by focusing on transductive node classification as a specific task, and then formulate the problem of pseudo-label selection in context of pseudo-labeling based SSGL.

**Semi-supervised graph learning (SSGL).** Consider a graph $\mathcal{G} = (\mathcal{V}, \mathcal{A}, \mathcal{X})$, where the node set $\mathcal{V} = \mathcal{V}_L \cup \mathcal{V}_U$ consists of a labeled node set $\mathcal{V}_L$ with labels $\mathcal{Y}_L = \{\boldsymbol{y}_v | v \in \mathcal{V}_L\}$ and an unlabeled node set $\mathcal{V}_U$. Each node $v \in \mathcal{V}$ belongs to one of $C$ categories and can be labeled with a $C$-dimensional one-hot vector $\boldsymbol{y}_v \in \mathbb{R}^C$. Here, $n_L = |\mathcal{V}_L|$ and $n_U = |\mathcal{V}_U|$ represent the number of nodes in the respective sets. In SSGL, only a limited number of labeled nodes are available and most nodes are unlabeled, *i.e.*, $n_L \ll n_U$. Adjacency matrix $\mathcal{A}$ characterizes edges connecting nodes in $\mathcal{V}$, and attribute matrix $\mathcal{X} \in \mathbb{R}^{(n_L+n_U) \times d_x}$ denotes node attribute, where $d_x$ denotes the dimension. SSGL aims to learn a parameterized model $f_\theta : v \rightarrow \hat{\boldsymbol{y}}_v, v \in \mathcal{V}_U$ that predicts the label of unlabeled nodes based on both the limited labeled nodes and other information in $\mathcal{G}$.

**Pseudo-labeling based SSGL**. In pseudo-labeling based SSGL, the label set $\mathcal{Y}_L$ is augmented with predicted pseudo-labels $\mathcal{Y}_P = \{\hat{\boldsymbol{y}}_v | v \in \mathcal{V}_P\}$, where the set $\mathcal{V}_P$ consists of unlabeled nodes with a high probability of being correctly predicted by the model $f_\theta$. Then, the model $f_\theta$ can be re-trained on the augmented label set, $\mathcal{Y}_L \cup \mathcal{Y}_P$, resulting in updated parameters $\theta^*$. In multi-stage pseudo-labeling based SSGL, the process of label augmentation and model retraining are repeated iteratively until convergence is achieved.

**Pseudo-label selection**. Pseudo-label selection is fundamental to pseudo-labeling based SSGL. It identifies unlabeled nodes that have high probabilities of being correctly predicted as pseudo-labeled nodes. Let $g_v \in \{0, 1\}$ be a binary indicator that signifies whether the node $v \in \mathcal{V}_U$ is selected for pseudo-labeling or not. Here, $g_v = 1$ when node $v$ is selected for pseudo-labeling, *i.e.*, $v$ is in $\mathcal{V}_P$, and $g_v = 0$ when $v$ is not selected. The pseudo-label selection is to obtain a indicator vector $\boldsymbol{g} = \{g_v | v \in \mathcal{V}_U\} \in \{0, 1\}^{n_U}$ for unlabeled nodes. This indicator can be obtained by a risk evaluation, $g_v = R_{f_\theta}(v) < \tau$, where $R_{f_\theta}(v)$ denotes generalization risk, *i.e.*, the probability of node $v$ being incorrectly predicted by the model $f_\theta$. A pseudo-labeled node can be identified if its generalization risk is less than the threshold $\tau$. However, the generalization risk is computationally intractable because real data distribution is unknown in practice. *Therefore, the pseudo-label selection becomes a problem of designing an effective measure that can estimate the generalization risk and act as a proxy of the risk for identifying pseudo-labeled nodes.*

## 3 METHOD

We first propose a novel uncertainty measure, memory disagreement MoDis, and then analyze the rationale of MoDis in case studies, and finally design MoDis based pseudo-labeling algorithm.

### 3.1 Memory Disagreement MoDis

The MoDis is proposed from the perspective of training dynamics and can serve as a proxy for the generalization risk to identify pseudo-labeled nodes in SSGL effectively. The computation of MoDis is illustrated in Fig. 1, in which we use training dynamics of three nodes in the Cora dataset as examples.

Firstly, we record the training dynamics for an unlabeled node, *i.e.,* the predictions made by model for the node at different training epochs, as shown in the left panel of Fig. 1. Next, we summarize the recorded training dynamics as a prediction distribution by calculating the relative frequency of categories in the predictions. Finally, we compute the entropy of the prediction distribution, which we define as the MoDis of the node, as shown in the right panel.

The proposed MoDis effectively captures the disagreement among predictions made by the model during training. This is because entropy quantifies the heterogeneity of the prediction distribution, and this heterogeneity reflects the diversity of predictions made by the model during training. The larger the disagreement among predictions, the higher the MoDis becomes. This relationship is clearly demonstrated by the comparison of the three nodes in Fig. 1.

**Training dynamics (predictions)**    **Memory disagreement**

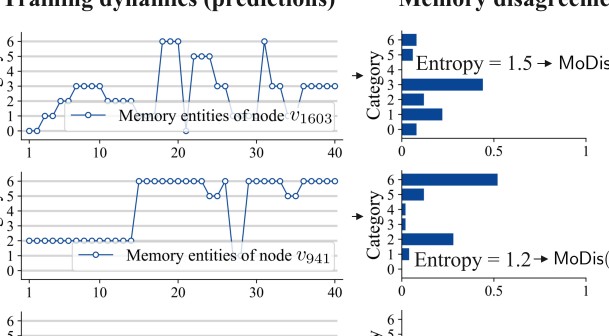

Figure 1: An illustration to compute memory disagreement MoDis. The left panel shows the prediction dynamics made by the model during training for three nodes in the Cora dataset. The right panel then summarizes these dynamics into distributions of predicted categories. The MoDis is defined as the entropy of the distributions.

Mathematically, the MoDis is defined as follows:

$$\text{MoDis}(v) := \mathbb{H}[P_v] = -\sum_{c=1}^{C} P_v(c) \log P_v(c), \quad (1)$$

where $\mathbb{H}[P_v]$ denotes the entropy of distribution $P_v$, and the $P_v$ is the distribution of predicted categories for node $v$. Each entry $P_v(c)$ is the relative frequency of category $c$ in multiple predictions during training. Specifically, $P_v(c)$ is calculated through

$$P_v(c) = \frac{1}{|\mathcal{M}_v|} \sum_{\hat{y}_v \in \mathcal{M}_v} \mathbb{1}[\hat{y}_v = c], \quad (2)$$

where the memory bank $\mathcal{M}_v$ records predicted labels $\hat{y}_v$ for each node $v$ at several selected training epochs, $|\mathcal{M}_v|$ denotes the size of the memory bank, and $\mathbb{1}[\cdot]$ is an indicator function used to verify whether predicted label $\hat{y}_v$ is equal to a certain category $c$ or not.

The memory bank $\mathcal{M}_v$ is leveraged to record training dynamics by storing representative predictions $\hat{y}$ at several specific epochs. Specifically, we define the memory bank as follows,

$$\mathcal{M}_v = \{\hat{y}_v^{(t)} | v \in \mathcal{V}_U, t \in \mathcal{T}\}. \quad (3)$$

Here $\hat{y}_v^{(t)}$ denotes the predicted label of node $v$ by the model $f_\theta$ at training epoch $t$, and set $\mathcal{T}$ contains the epochs selected to record. During training, we begin with an empty $\mathcal{M}_v$ for each node and progressively add the predicted labels $\hat{y}_v^{(t)}$ at selected epochs in $\mathcal{T}$.

For better practicality in real-world applications, we streamline the epoch set $\mathcal{T}$ into a sequence of consecutive epochs during training. This sequence can be easily determined using two hyper-parameters: the starting epoch for recording and the number of epochs to be recorded. In experiments, we empirically analyze how to construct an effective $\mathcal{M}$ by the two hyper-parameters. This simplifying method can capture the primary trajectory of predictions throughout training, with minimized computational and storage overhead, enabling the efficient computation of MoDis.

## 3.2 Enhancing MoDis via Softmax Trajectory

We further enhance memory disagreement by incorporating the trajectory of softmax distribution of predictions during training. In preliminary studies, we observe a resolution limitation of MoDis defined in Eq. (1). That is, different nodes sometimes get the same accumulated distribution of predicted categories, and then they have identical MoDis values, rendering them indistinguishable.

To improve the discriminative ability of MoDis, instead of using the distribution of predicted categories in Eq. (2), we refine $P_v$ by accumulating the softmax distributions during training as

$$P_v(c) = \frac{1}{|\mathcal{M}_v|} \sum_{\sigma_v \in \mathcal{M}_v} \sigma_v(c), \quad (4)$$

where the memory bank $\mathcal{M}_v$ is used to record softmax distributions $\sigma_v$ of prediction during training. For a selected epoch $t \in \mathcal{T}$, $\sigma_v^{(t)} = \text{softmax}(z_v^{(t)})$ represents the softmax distribution with $z_v^{(t)}$ being the logits of node $v$. This refined $P_v$ is an average of the softmax distributions, which characterizes subtle variances in predictions and captures richer information of training dynamics. Although softmax distribution is not a reliable estimator of prediction uncertainty, it can be considered as data uncertainty [12].

Additionally, we introduce a sharpening step to the softmax distributions inspired by the efficacy of entropy minimization in SSL [3]. Given the softmax distributions, we apply a sharpening function to enhance the decisiveness of the distribution by reducing its entropy, which is achieved through a sharpening function,

$$\text{Sharpen}(\boldsymbol{p}, \gamma) := \boldsymbol{p}_i^{1/\gamma}/Z, \ Z = \sum_{l=1}^{L} \boldsymbol{p}_l^{1/\gamma}.$$

Here, the sharpening function is defined on each category. Vector $\boldsymbol{p}$ denotes an $L$-dimensional categorical distribution, which is the predicted softmax distribution $\sigma_v$ in the context of memory disagreement. The "temperature" hyperparameter $\gamma$ is used to adjust the shape of the distribution. For $\gamma = 1$, the sharpening function retains the original shape of the distribution. As $\gamma \rightarrow 0$, the result of Sharpen$(\boldsymbol{p}, \gamma)$ becomes a one-hot distribution, making the $P_v$ in Eq. (2) degraded to its form in Eq. (4). The sharpening function amplifies more probable predictions while decaying less probable ones, thereby emphasizing confident predictions more strongly.

## 3.3 Case Study Analysis of MoDis

We design two case studies to demonstrate the characteristics of nodes with a low MoDis value, as well as analyze the rationale of these nodes as candidates for pseudo-labeling in SSGL. We visualize these nodes in both the graph and representation spaces for an intuitive understanding. A notable observation is that these nodes tend to be distant from two kinds of boundaries.

In graph space, nodes with low MoDis predominantly are not positioned on the boundary. For clarification, we define boundary nodes in the graph as nodes whose labels differ from those of their neighbors, as illustrated in Fig. 2(A). We conducted an analysis on five graph datasets. In this study, we compute the proportion of boundary nodes in three groups: the top 100 nodes with the lowest MoDis, those with the highest MoDis, and overall nodes in graph. The proportion of boundary nodes was significantly lower among nodes with low MoDis when compared to those with high MoDis, as shown in Fig. 2(B). Nodes with a low MoDis are more internally

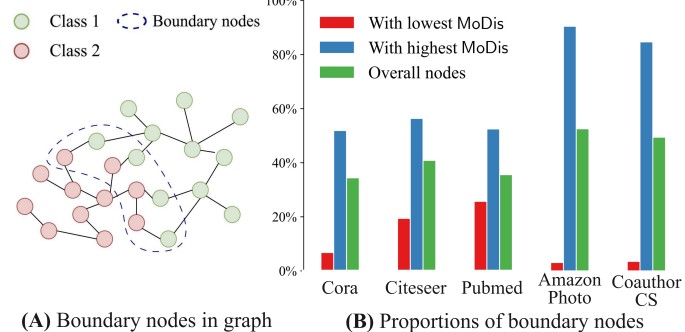

**(A)** Boundary nodes in graph    **(B)** Proportions of boundary nodes

**Figure 2: In graph space, nodes with low** MoDis **predominantly are not positioned on the boundary. (A) An illustration of boundary nodes in graph. (B) Comparative proportions of boundary nodes of three different node groups.**

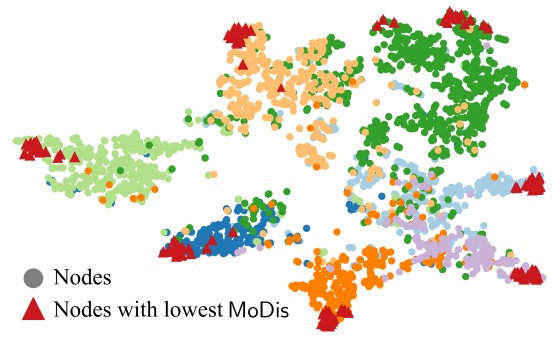

● Nodes

▲ Nodes with lowest MoDis

**Figure 3: In representation space, nodes with low** MoDis **cluster in areas distanced from decision boundaries. Node representations are visualized in a 2D space. Red triangles indicate the top 20 nodes in each category with the lowest** MoDis.

consistent with their neighbors. *This consistency suggests they are more reliable candidates for pseudo-labeling, as their representations are less likely to be mixed with irrelevant messages from nodes with different label during the aggregation in GNNs.*

In representation space, nodes with low MoDis exhibit spatial clustering at positions removed from decision boundaries, as shown in Fig. 3. Here, we visualize node representations in a 2D space, which are extracted by the last layer of GCN [18] on the Cora dataset, and further mapped using t-SNE [36]. The nodes with low MoDis are distant from decision boundaries, which implies they are a safe choice as pseudo-labeling, as they have a relatively low risk of being incorrectly predicted. *These nodes distanced from decision boundaries are safer candidates for pseudo-labeling, as they have a lower risk of misclassification, ensuring minimal noise introduction to the model in pseudo-labeling.*

### 3.4 MoDis **based Pseudo-labeling Algorithm**

In this section, we first propose a MoDis based pseudo-label selection algorithm, as illustrated in Algorithm 1. We then incorporate the pseudo-label selection algorithm into a pseudo-labeling framework for SSGL, as presented in Algorithm 2.

---

**Algorithm 1** MoDis based Pseudo-label Selection

---

**Input:** Graph $\mathcal{G} = (\mathcal{V}, \mathcal{A}, \mathcal{X})$, label set $\mathcal{Y}_L$, graph model $f_\theta$
**Output:** Pseudo-label set $\mathcal{Y}_P$
**Parameter:** Epoch index set $\mathcal{T}$

1: Initialize model parameters $\theta$, memory bank $\mathcal{M}_v \leftarrow \varnothing$ for each node $v \in \mathcal{V}_U$, and $\mathcal{Y}_P \leftarrow \varnothing$
2: **for** training epoch $t = 1$ to max_epoch **do**
3:     Update $\theta$ using the gradient calculated on $\mathcal{Y}_L$
4:     **if** $t \in \mathcal{T}$ **then**
5:         **for** each node $v \in \mathcal{V}_U$ **do**
6:             Compute $\sigma_v^{(t)}$ by model $f_\theta(\mathcal{A}, \boldsymbol{x}_v)$
7:             $\mathcal{M}_v \leftarrow \mathcal{M}_v \cup \{\sigma_v^{(t)}\}$
8: **for** each node $v \in \mathcal{V}_U$ **do**
9:     Calculate MoDis$(v)$ using Eq. (1) and (4)
10:     **if** MoDis$(v) < \tau$ **then**
11:         $\hat{\boldsymbol{y}}_v = f_\theta(\mathcal{A}, \boldsymbol{x}_v)$; $\mathcal{Y}_P \leftarrow \mathcal{Y}_P \cup \{\hat{\boldsymbol{y}}_v\}$
12: **return** $\mathcal{Y}_P$

---

In Algorithm 1, a training process for the graph neural network $f_\theta$ is performed on graph data $\mathcal{G}$ firstly. Throughout the training, we use a memory bank $\mathcal{M}$ to capture the softmax distributions of predictions at specific epochs, $t \in \mathcal{T}$. Subsequently, the recorded training dynamics are used to calculate MoDis for every unlabeled node using Eq. (1) and (4). Finally, unlabeled nodes with a MoDis value below the threshold $\tau$ are selected as pseudo-labeled nodes. The output pseudo-labels are predicted by the fully-trained model, and they are then used to augment the limited real labels in $\mathcal{G}$.

In Algorithm 2, we present MoDis pseudo-labeling based SSGL by incorporating Algorithm 1 into a multi-stage self-training framework. This algorithm consists of $K$ stages. Within each stage, pseudo labels $\mathcal{Y}_P$ are first produced by Algorithm 1 leveraging currently available labels in $\mathcal{Y}_L$. Following this, the label set $\mathcal{Y}_L$ is augmented by these pseudo labels, and the augmented label set serves as the foundation for the next round of pseudo-label generation. To avoid the issue of label imbalance, we add an equal number of pseudo labels for each category in every stage.

After iterating through the $K$ stages, the final model is trained using the ultimately augmented label set $\mathcal{Y}_L$, and is used to predict labels for all unlabeled nodes in the graph $\mathcal{G}$. Notably, for unlabeled nodes that have previously been associated with pseudo labels in the $K$ stages, those pseudo labels are retained as their final predictions.

---

**Algorithm 2** MoDis Pseudo-labeling based SSGL

---

**Input:** Graph $\mathcal{G} = (\mathcal{V}, \mathcal{A}, \mathcal{X})$, label set $\mathcal{Y}_L$, graph model $f_\theta$
**Output:** Predicted label set $\mathcal{Y}^*$ for all unlabeled nodes
**Parameter:** Number of stage $K$

1: Initialize $\mathcal{Y}^* \leftarrow \varnothing$
2: **for** each stage $k = 1$ to $K$ **do**
3:     $\mathcal{Y}_P$ = Algorithm 1$(\mathcal{G}, \mathcal{Y}_L, f_\theta)$
4:     $\mathcal{Y}_L \leftarrow \mathcal{Y}_L \cup \mathcal{Y}_P$; $\mathcal{Y}^* \leftarrow \mathcal{Y}^* \cup \mathcal{Y}_P$
5:     Update $\mathcal{V}_L$ and $\mathcal{V}_U$ in $\mathcal{G}$ according to $\mathcal{Y}_L$
6:     Train model $f_\theta$ on augmented $\mathcal{Y}_L$
7:     **for** each node $v \in \mathcal{V}_U$ **do**
8:         $\hat{\boldsymbol{y}}_v = f_\theta(\mathcal{A}, \boldsymbol{x}_v)$; $\mathcal{Y}^* \leftarrow \mathcal{Y}^* \cup \{\hat{\boldsymbol{y}}_v\}$
9: **return** $\mathcal{Y}^*$

---

**Table 1: Dateset statistics**

| Dataset | # Nodes | # Edges | # Categories | # Features |
|---|---|---|---|---|
| Cora [28] | 2,708 | 5,429 | 7 | 1,433 |
| Citeseer [28] | 3,327 | 4,732 | 6 | 3,703 |
| Pubmed [23] | 19,717 | 44,338 | 3 | 500 |
| CoraFull [4] | 19,793 | 65,331 | 70 | 8,710 |
| AmazonCS [30] | 13,752 | 245,778 | 10 | 767 |
| AmazonPhoto [30] | 7,650 | 119,043 | 8 | 745 |
| CoauthorCS [30] | 18,333 | 81,894 | 15 | 6,805 |
| CoauthorPhy. [30] | 34,493 | 495,924 | 5 | 8,415 |

## 4 EXPERIMENTS

We empirically validate the effectiveness of our proposed MoDis by answering the following four questions.

**Q1**: Does MoDis outperform competitors (state-of-the-art uncertainty measures) in selecting pseudo-labeled nodes?

**Q2**: Does graph neural networks equipped with MoDis pseudo-labeling outperform that with competitors in SSGL?

**Q3**: Is MoDis effective for out-of-distribution nodes?

**Q4**: How does the training dynamics, captured in the memory bank $\mathcal{M}$, affect performance of MoDis based pseudo-labeling?

**Baselines.** We choose the following two state-of-the-art baselines as competitors of MoDis in experiments.

- *Confidence score*: The most frequently used uncertainty measure for selecting pseudo-labeled nodes in SSGL [5, 15].
- *Area Under the Margin (AUM)*: A recent measure that has been successfully applied to semi-supervised learning in natural language processing [33].

**Datasets.** We conducted experiments on eight benchmark graph datasets, namely, Cora, Citeseer [28], Pubmed [23], CoraFull [4], AmazonComputers, AmazonPhoto, CoauthorCS, and Coauthor-Physics [30]. The statistics of the datasets are summarized in Table 1. The detailed experiment protocol can be found in Appendix.

## 4.1 Quality of Pseudo Labels (Q1)

We first examine whether the proposed MoDis can select better pseudo-labeled nodes than the competitors. We evaluate the quality of pseudo labels from two perspectives: correctness and information gain. This experiment is also designed to validate our selected pseudo-labeled nodes contained simple patterns (high correctness) and general patterns (large information gain). We employ GCN as the base model $f_\theta$, following hyper-parameter settings in [18]. To simulate a challenging scenario with minimal label information, we only set 3 labeled nodes per category, *i.e.*, $L/C = 3$, in six graphs. Pseudo labels are produced by Algorithm 1 or its variants in which uncertainty measure MoDis is substituted with the competitors.

*4.1.1 Correctness of Pseudo Labels.* We first evaluate the correctness of produced pseudo labels in terms of *error ratio*. Specifically, the error ratio is defined as the proportion of nodes for which the pseudo label and true label do not match, in entire pseudo-labeled nodes. We generate pseudo labels for all unlabeled nodes in a graph, subsequently sorting these nodes based on the three uncertainty measures. The nodes are then orderly partitioned into 10 evenly groups, and independently computed the error ratio for each group.

Fig. 4 shows the error ratio for the first 5 groups, as these nodes are more likely to be selected for pseudo-labeling in practice. From

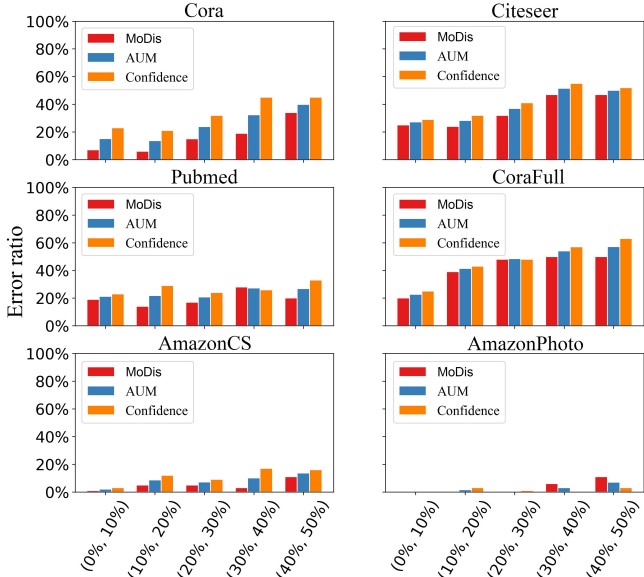

**Figure 4: The error ratios of pseudo-labeled nodes on six graphs. The proposed MoDis selects pseudo-labeled nodes with the lowest error ratios in most scenarios compared to confidence and AUM. The $x$-axis denotes node groups that imply priority of nodes being selected for pseudo-labeling.**

the results, we make two observations: (i) The error ratio consistently increases as we progress through successive groups, which corresponds to the order of pseudo-labeled nodes. This increasing trend indicates the efficacy of the 3 uncertainty measures: lower uncertainty correlates with higher correctness. (ii) our proposed MoDis outperforms confidence score and AUM in most scenarios, providing pseudo-labeled nodes with the lowest error ratios.

*4.1.2 Information Gain from Pseudo Labels.* We also evaluate the information gained from pseudo labels to the model. Base on the concept of expected gradient length in active learning [29], we design model perturbation $\rho$ to measure the information gain, which is the change in the gradient caused by adding pseudo labels,

$$\rho = \|\nabla\ell\left(\mathcal{Y}_L \cup \mathcal{Y}_P; f_\theta\right)\|_F - \|\nabla\ell\left(\mathcal{Y}_L; f_\theta\right)\|_F,$$

where $\nabla\ell(\cdot)$ denotes the gradient of model $f_\theta$, $\mathcal{Y}_L$ and $\mathcal{Y}_P$ denote real label set and pseudo label set, respectively. Frobenius norm $\|\cdot\|_F$ is used to quantify the gradient. For simplicity, we first train the model using labels in $\mathcal{Y}_L$ until it converges. As a result, the value of $\|\nabla\ell(\mathcal{Y}_L; \theta)\|_F$ approaches zero, which allows us to approximate

**Table 2: The model perturbation caused by pseudo labels**

| CPL/C | 20 | 30 | 20 | 30 | 20 | 30 |
|---|---|---|---|---|---|---|
| Dataset | Cora | | Citeseer | | Pubmed | |
| Confidence | 181.1 | 278.4 | 157.2 | 235.7 | 45.4 | 68.5 |
| AUM | 179.9 | 272.1 | 155.9 | 240.8 | 39.8 | 55.9 |
| MoDis | **187.3** | **302.2** | **172.2** | **257.8** | **45.4** | **68.5** |
| Dataset | CoraFull | | AmazonCS | | AmazonPhoto | |
| Confidence | 4568 | 6010 | 1742 | 2528 | 1320 | 1880 |
| AUM | 4492 | 5926 | 1652 | 2470 | 1224 | 1795 |
| MoDis | **4869** | **6797** | **2108** | **5869** | **1392** | **1970** |

**Table 3: The node classification accuracy of pseudo-labeling algorithms on citation graphs (%)**

| Dataset | Cora | | | | Citeseer | | | | Pubmed | | | | CoraFull | | | |
|---|---|---|---|---|---|---|---|---|---|---|---|---|---|---|---|---|
| L/C | 3 | 5 | 10 | 20 | 3 | 5 | 10 | 20 | 3 | 5 | 10 | 20 | 3 | 5 | 10 | 20 |
| Pseudo-labeling algorithms with GCN | | | | | | | | | | | | | | | | |
| Base model | 64.47 | 74.06 | 77.85 | 80.88 | 58.04 | 63.73 | 70.84 | 72.02 | 67.93 | 71.52 | 74.98 | 79.78 | 41.83 | 49.90 | 56.20 | 62.07 |
| Confidence | 63.09 | 72.51 | 77.01 | 80.01 | 60.22 | 64.41 | 69.39 | 71.57 | 63.41 | 71.46 | 72.01 | 77.90 | 43.66 | 50.72 | 56.93 | 61.64 |
| AUM | 64.28 | 71.84 | 76.74 | 79.89 | 58.99 | 64.99 | 70.13 | 71.42 | 63.51 | 70.84 | 70.51 | 77.00 | 43.85 | 51.14 | 57.46 | 61.86 |
| MoDis | 64.92 | 74.52 | 79.29 | 81.00 | 59.89 | 65.16 | 69.97 | 71.75 | 67.87 | 71.89 | 74.90 | 79.73 | 43.97 | 51.21 | 57.68 | 61.88 |
| Confidence-MS | 63.09 | 72.47 | 76.95 | 79.64 | 62.10 | 65.65 | 71.04 | 72.31 | 60.19 | 67.01 | 70.04 | 76.86 | 42.77 | 51.26 | 56.98 | 61.53 |
| AUM-MS | 63.88 | 72.03 | 76.98 | 79.63 | 61.90 | 66.09 | 71.05 | **72.44** | 59.55 | 66.35 | 69.58 | 73.72 | 42.16 | 51.16 | 56.87 | 61.54 |
| MoDis-MS | **69.00** | **75.36** | **79.84** | **81.58** | **65.24** | **68.59** | **71.19** | 72.37 | **70.32** | **72.54** | **75.05** | **79.84** | **44.70** | **51.30** | **57.93** | **62.26** |
| Pseudo-labeling algorithms with GAT | | | | | | | | | | | | | | | | |
| Base model | 63.84 | 70.38 | 78.65 | 81.78 | 43.77 | 56.94 | 67.91 | 70.65 | 66.45 | 68.94 | **74.63** | **77.39** | 39.17 | 47.09 | 53.12 | 59.65 |
| Confidence | 73.99 | 74.67 | 78.34 | 81.42 | 64.02 | 67.27 | 69.15 | 70.65 | 64.52 | 65.68 | 69.10 | 72.05 | 39.90 | 48.22 | 55.57 | 60.06 |
| AUM | 67.77 | 75.01 | 78.63 | 80.94 | 61.13 | 66.62 | 70.15 | 71.11 | 67.34 | 68.51 | 70.10 | 72.57 | 40.01 | 49.40 | 54.97 | 60.56 |
| MoDis | 72.90 | 74.72 | 78.93 | 81.54 | 59.31 | 67.52 | 70.25 | 71.50 | 68.20 | 69.16 | 71.98 | 74.11 | 41.11 | 49.07 | 54.58 | 60.23 |
| Confidence-MS | 73.24 | 75.13 | 78.34 | 81.42 | 64.02 | 67.27 | 69.15 | 70.65 | 64.52 | 65.68 | 69.10 | 72.05 | 39.34 | 48.30 | 54.83 | 60.16 |
| AUM-MS | 67.18 | 74.52 | 78.53 | 79.64 | 63.15 | 66.99 | 69.45 | 70.40 | 64.69 | 67.75 | 67.53 | 70.01 | 38.67 | 48.83 | 54.33 | 59.96 |
| MoDis-MS | **75.17** | **75.90** | **78.95** | **82.05** | **65.22** | **67.90** | **70.97** | **72.56** | **68.86** | **70.05** | 72.82 | 76.63 | **43.82** | **50.72** | **55.57** | **61.09** |
| Pseudo-labeling algorithms with APPNP | | | | | | | | | | | | | | | | |
| Base model | 65.66 | 74.05 | 79.76 | 81.99 | 44.49 | 60.29 | 66.54 | 70.69 | 66.83 | 72.29 | 75.59 | 78.78 | 40.74 | 48.94 | 55.37 | **61.01** |
| Confidence | 73.39 | 76.70 | 80.56 | 81.86 | 60.63 | 68.51 | 71.28 | 71.98 | 69.40 | 71.35 | 75.80 | 78.36 | 43.05 | 49.96 | 55.51 | 58.46 |
| AUM | 71.35 | 77.50 | 80.03 | 81.94 | 59.50 | 67.30 | 70.44 | 71.84 | 69.73 | 72.31 | 74.21 | 76.66 | 43.09 | 49.92 | 56.17 | 59.65 |
| MoDis | 74.26 | 76.56 | 80.00 | 82.10 | 61.04 | 66.71 | 71.52 | 72.45 | 70.14 | 71.71 | 76.79 | 79.21 | 42.68 | 48.49 | 54.62 | 59.62 |
| Confidence-MS | 73.48 | 76.69 | 80.76 | 82.08 | 63.74 | 66.46 | 71.22 | 71.31 | 69.34 | 69.35 | 73.23 | 74.49 | 40.96 | 48.52 | 54.55 | 57.85 |
| AUM-MS | 70.11 | 76.72 | 80.70 | 82.05 | 64.22 | 69.23 | 71.30 | 71.30 | 69.55 | 69.91 | 70.84 | 71.80 | 40.48 | 49.12 | 54.69 | 59.37 |
| MoDis-MS | **75.85** | **77.79** | **80.95** | **83.32** | **65.13** | **69.62** | **71.58** | **72.70** | **70.23** | **72.39** | **77.02** | **79.37** | **43.23** | **50.15** | **56.18** | 60.00 |
| Pseudo-labeling algorithms with GCNII | | | | | | | | | | | | | | | | |
| Base model | 64.61 | 71.25 | 78.62 | 84.75 | 50.83 | 61.57 | 69.23 | 72.69 | 64.35 | 70.55 | 75.96 | 78.39 | 43.08 | 49.35 | 56.52 | 61.08 |
| Confidence | 65.11 | 70.90 | 79.60 | 84.45 | 51.69 | 61.24 | 69.40 | 72.26 | 67.79 | 68.94 | 74.09 | 78.44 | 41.63 | 47.59 | 55.03 | 59.30 |
| AUM | 63.22 | 73.34 | 79.75 | 84.45 | 53.24 | 56.37 | 68.98 | 72.21 | **69.28** | 71.06 | 74.01 | 78.50 | 41.56 | 48.62 | 56.46 | 61.22 |
| MoDis | 65.11 | 75.94 | 80.04 | 84.45 | 55.06 | 61.24 | 69.40 | 72.26 | 71.35 | 72.22 | 74.09 | 78.50 | 42.45 | 49.90 | 57.15 | 61.22 |
| Confidence-MS | 63.55 | 71.75 | 79.88 | 84.52 | 49.96 | 60.07 | 70.25 | 71.68 | 66.87 | 69.54 | 75.86 | 79.25 | 41.93 | 48.87 | 55.31 | 59.56 |
| AUM-MS | 63.06 | 74.65 | 79.90 | 84.65 | 51.94 | 57.68 | 68.24 | 72.75 | 67.92 | 70.65 | 71.67 | 79.38 | 42.64 | 49.40 | 56.78 | 61.07 |
| MoDis-MS | **77.55** | **78.14** | **80.45** | **85.08** | **65.07** | **68.54** | **71.43** | **72.86** | **69.28** | 71.31 | 76.13 | 79.43 | **43.26** | **50.28** | **57.68** | **61.57** |

model perturbation $\rho$ as $\|\nabla \ell\left(\mathcal{Y}_P; \theta\right)\|_F$ directly. Here, the model perturbation $\rho$ is calculated on a given number of correct pseudo-labeled nodes produced. Only correct pseudo labels are used, as what incorrect pseudo labels introduce is harmful information. CPL/C denotes the number of correct pseudo labels per class.

Table 2 shows the model perturbation caused by adding pseudo-labeled nodes given by the three uncertainty measures. It can be observed that in all scenarios, pseudo-labeled nodes given by MoDis cause greater perturbations on gradient, which suggests the model gains more information from these pseudo labels.

From the evaluation of correctness and information gain, it can be concluded that *our proposed MoDis outperforms confidence score and AUM, as the pseudo-labeled nodes slected by MoDis are not only* **more correct** *but also* **more informative**.

## 4.2 Comparisons on Node Classification (Q2)

We further validate the efficacy of our proposed MoDis by evaluating the performance of graph neural networks equipped with MoDis based pseudo-labeling. The experiments are conducted on eight benchmark graph datasets on inductive node classification tasks. We adopt classification accuracy as evaluation metric.

Node labels are predicted by using Algorithm 2, and pseudo labels are produced by Algorithm 1 or its variants in which MoDis is replaced with confidence score or AUM. We adopt four widely-used graph neural networks, GCN [18], GAT [37], APPNP[11] and GCNII [7], as the base model $f_\theta$ in Algorithm 2, following the same hyper-parameter settings in their original papers.

The extensive comparative results are summarized in Table 3 and 4. In these tables, "Base model" denotes the corresponding base model without pseudo-labeling; "Confidence", "AUM", and "MoDis" represent the base model equiped with respective pseudo-labeling methods; an "MS" suffix indicates a multi-stage pseudo-labeling, *i.e.*, $K > 1$ in Algorithm 2. In the absence of this suffix, $K = 1$.

The results presented in Table 3 and 4 highlight that our proposed MoDis consistently outperforms the competitors in most scenarios. Specifically, we have the following three observations:

- The performance of GCN, GAT, APPNP, and GCNII models decreases significantly as the number of labeled nodes per category (L/C) decreases. This indicates that pseudo-labeling algorithms bring performance improvement compared to base models, especially when the number of labeled nodes is very limited.

**Table 4: The node classification accuracy of pseudo-labeling algorithms on citation and co-purchasing graphs (%)**

| Dataset | AmazonCS | | | | AmazonPhoto | | | | CoauthorCS | | | | CoauthorPhysics | | | |
|---|---|---|---|---|---|---|---|---|---|---|---|---|---|---|---|---|
| L/C | 3 | 5 | 10 | 20 | 3 | 5 | 10 | 20 | 3 | 5 | 10 | 20 | 3 | 5 | 10 | 20 |
| Pseudo-labeling algorithms with GCN | | | | | | | | | | | | | | | | |
| Base model | 66.59 | 76.68 | 80.41 | 82.80 | 87.08 | 87.53 | 91.39 | 93.42 | 88.83 | 90.72 | 91.52 | 92.34 | 84.20 | 85.59 | 88.50 | 89.78 |
| Confidence | 67.18 | 77.26 | 80.58 | 82.43 | 86.60 | 88.67 | 91.23 | 93.21 | 87.75 | 90.71 | 92.39 | 92.84 | 85.39 | 86.93 | 88.75 | 90.10 |
| AUM | 68.25 | 77.01 | 80.27 | 82.64 | 86.61 | 88.59 | 91.14 | 93.56 | 87.99 | 90.68 | 92.32 | 92.86 | 85.86 | 87.15 | 88.54 | 90.00 |
| MoDis | 68.72 | 76.74 | 81.00 | 82.76 | 87.30 | 88.19 | 91.68 | 93.62 | 89.01 | 90.68 | 92.31 | 92.84 | 85.65 | 87.35 | 88.87 | 90.10 |
| Confidence-MS | 66.89 | 77.08 | 80.72 | 82.01 | 86.41 | 88.21 | 91.15 | 93.20 | 87.08 | 90.83 | 92.10 | 92.72 | 85.75 | 87.36 | 88.81 | 90.13 |
| AUM-MS | 66.54 | 78.38 | 80.84 | 82.34 | 85.85 | 88.42 | 90.91 | 93.12 | 87.94 | 90.94 | 92.11 | 92.84 | 85.71 | 87.40 | 88.65 | 90.11 |
| MoDis-MS | 68.97 | 77.37 | 81.19 | 83.06 | 88.03 | 89.23 | 91.87 | 93.79 | 89.91 | 92.54 | 93.34 | 93.62 | 86.46 | 88.00 | 91.00 | 92.36 |
| Pseudo-labeling algorithms with GAT | | | | | | | | | | | | | | | | |
| Base model | 63.80 | 72.09 | 80.59 | 82.45 | 81.27 | 89.33 | 91.98 | 92.58 | 90.60 | 91.34 | 92.25 | 92.66 | 80.64 | 82.70 | 88.82 | 89.89 |
| Confidence | 63.84 | 75.79 | 80.64 | 82.03 | 87.11 | 88.74 | 91.27 | 92.91 | 88.30 | 90.79 | 92.10 | 92.81 | 83.44 | 85.74 | 88.04 | 88.68 |
| AUM | 64.58 | 75.70 | 79.73 | 81.72 | 88.79 | 87.55 | 92.10 | 93.10 | 89.35 | 90.60 | 92.26 | 92.89 | 84.84 | 85.84 | 88.47 | 89.83 |
| MoDis | 67.99 | 76.85 | 81.15 | 82.36 | 87.99 | 89.12 | 91.79 | 93.27 | 90.35 | 90.72 | 92.35 | 92.94 | 83.30 | 85.89 | 88.44 | 90.11 |
| Confidence-MS | 66.75 | 75.46 | 79.67 | 81.99 | 87.66 | 89.11 | 91.20 | 93.04 | 90.01 | 91.09 | 91.65 | 92.86 | 83.01 | 86.20 | 87.78 | 89.88 |
| AUM-MS | 67.68 | 75.72 | 79.78 | 82.19 | 87.59 | 88.81 | 91.94 | 93.29 | 89.86 | 91.29 | 92.24 | 92.91 | 83.62 | 87.32 | 87.84 | 89.70 |
| MoDis-MS | 70.85 | 78.91 | 81.16 | 82.52 | 88.19 | 89.45 | 92.06 | 93.46 | 91.16 | 91.54 | 92.57 | 93.09 | 87.97 | 88.89 | 89.10 | 90.30 |
| Pseudo-labeling algorithms with APPNP | | | | | | | | | | | | | | | | |
| Base model | 68.50 | 76.60 | 80.45 | 81.78 | 87.04 | 88.78 | 91.43 | 93.52 | 89.85 | 90.97 | 91.94 | 92.75 | 86.18 | 87.52 | 88.34 | 90.03 |
| Confidence | 68.67 | 76.37 | 78.29 | 81.31 | 87.10 | 89.27 | 92.26 | 93.82 | 87.65 | 90.80 | 92.23 | 93.03 | 87.83 | 88.41 | 89.63 | 90.95 |
| AUM | 68.73 | 75.59 | 79.93 | 81.45 | 87.74 | 89.53 | 92.26 | 93.87 | 89.95 | 91.10 | 92.52 | 93.11 | 87.41 | 88.47 | 89.64 | 90.63 |
| MoDis | 70.09 | 76.31 | 80.63 | 82.33 | 87.73 | 89.65 | 92.43 | 93.92 | 91.39 | 92.11 | 92.72 | 93.16 | 87.69 | 89.53 | 89.87 | 91.05 |
| Confidence-MS | 65.17 | 75.96 | 80.18 | 80.99 | 86.14 | 89.96 | 92.19 | 93.45 | 87.33 | 91.14 | 91.87 | 92.68 | 87.74 | 88.25 | 89.60 | 90.81 |
| AUM-MS | 66.14 | 75.18 | 79.94 | 81.44 | 87.75 | 88.54 | 91.77 | 93.51 | 89.50 | 91.12 | 92.47 | 92.95 | 86.78 | 87.63 | 89.22 | 90.19 |
| MoDis-MS | 70.26 | 78.58 | 81.12 | 82.46 | 89.03 | 90.04 | 92.52 | 93.97 | 92.04 | 92.18 | 92.79 | 93.28 | 88.32 | 89.62 | 89.95 | 91.32 |
| Pseudo-labeling algorithms with GCNII | | | | | | | | | | | | | | | | |
| Base model | 62.92 | 74.07 | 78.86 | 80.87 | 84.84 | 86.58 | 91.78 | 92.85 | 91.26 | 91.57 | 93.19 | 94.00 | 87.94 | 88.96 | 89.62 | 90.88 |
| Confidence | 61.30 | 73.94 | 76.88 | 79.26 | 82.71 | 84.46 | 91.05 | 91.71 | 90.38 | 91.50 | 93.11 | 94.06 | 88.36 | 88.31 | 89.51 | 91.34 |
| AUM | 61.56 | 74.16 | 77.79 | 79.94 | 83.32 | 85.67 | 91.65 | 92.96 | 90.87 | 91.89 | 93.26 | 94.06 | 87.92 | 88.41 | 89.20 | 91.03 |
| MoDis | 62.66 | 74.57 | 78.81 | 80.19 | 84.77 | 86.32 | 91.93 | 93.13 | 91.39 | 92.38 | 93.33 | 94.09 | 88.45 | 88.65 | 89.81 | 91.34 |
| Confidence-MS | 63.74 | 69.69 | 77.27 | 78.95 | 82.87 | 85.13 | 91.44 | 91.19 | 90.61 | 91.53 | 92.44 | 93.66 | 88.37 | 89.39 | 90.00 | 91.80 |
| AUM-MS | 62.00 | 76.38 | 78.61 | 79.96 | 84.81 | 86.36 | 92.35 | 93.21 | 91.62 | 92.00 | 93.04 | 93.82 | 88.28 | 88.15 | 90.33 | 91.43 |
| MoDis-MS | 64.52 | 76.67 | 78.95 | 81.02 | 85.06 | 86.79 | 92.83 | 93.54 | 91.95 | 92.42 | 93.83 | 94.55 | 89.91 | 90.89 | 91.05 | 92.27 |

- Employing MoDis based pseudo-labeling algorithm always results in better classification accuracy than competitors, compared to the wildly-used confidence score, achieving an average improvement of 3.11% (the average performance gap between MoDis-MS and Confidence-MS in all columns of Table 3 and 4) and reaching up to 30.24% (L/C = 3, GCNII model, Citeseer dataset). This again demonstrates the efficacy of MoDis.
- Multi-stage pseudo-labeling algorithms offer significant benefits when working with very limited labeled nodes. However, the benefits seem to reduce as the quantity of labeled nodes increases.

## 4.3 Validation on OOD Nodes (Q3)

In this section, we validate the proposed MoDis in handling out-of-distribution (OOD) nodes. We integrate the proposed pseudo-labeling algorithm with OODGAT [32] that is a model specifically designed for OOD tasks in graphs. We adopt the same experimental setup in the original paper, except for the number of labeled nodes per category $L/C = 3$. We divide nodes into in-distribution and out-of-distribution classes, as detailed in Appendix.

This validation consists of two tasks, in-distribution node classification and out-of-distribution detection. Two metrics are adopted

**Table 5: The experimental results in OOD setting**

| Dataset | Cora | AmazonCS | AmazonPhoto | CoauthorCS |
|---|---|---|---|---|
| Metric | ACC↑ (in-distribution) / FPR@95↓ (out-of-distribution) | | | |
| Base model | 74.2/64.7 | 68.7/56.9 | 93.5/25.5 | 70.7/14.2 |
| Confidence | 79.1/53.6 | 71.2/54.6 | 95.5/27.5 | 70.8/21.5 |
| AUM | 80.1/48.5 | 71.8/54.5 | 95.2/26.8 | 73.3/**13.6** |
| MoDis | **82.1/43.4** | **73.9/52.9** | **97.3/18.8** | **76.5**/14.7 |

to evaluate the two tasks, respectively. Specifically, classification accuracy (ACC) is used on node classification, and false positive rate at 95% true positive rate (FPR@95) are used on out-of-distribution detection. In experiments, node labels are predicted by using Algorithm 2, in which we adopt OODGAT-ATT as the base model $f_\theta$. Pseudo labels are produced by Algorithm 1 or its variants in which MoDis is replaced with confidence score or AUM.

The experimental results in OOD setting are summarized in Table 5. We can observe that the model OODGAT-ATT equipped with MoDis based pseudo-labeling algorithm markedly improves the accuracy of in-distribution node classification on the four datasets.

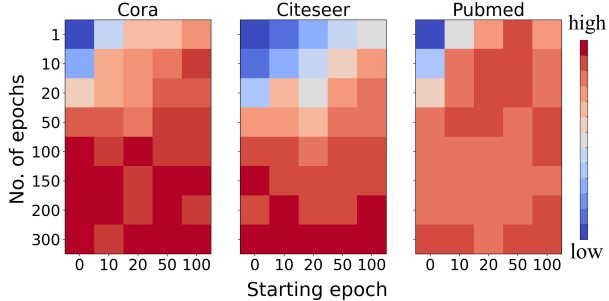

**Figure 5: The results of grid search — accuracies of node classification in different settings of the two hyper-parameters.**

Remarkably, MoDis also demonstrate efficacy in detecting out-of-distribution nodes, outperforming competitors in most scenarios.

We also observe confidence score achieves improvements. This likely be attributed to the high homophily in the graphs [26]. The graph neural network, owing to the inherent smoothing effect of propagation, tends to be more confident for ID nodes.

### 4.4 Analysis on Training Dynamics (Q4)

We further investigate memory bank, which is fundamental in the proposed method and is used to captures training dynamics. To this end, we execute a grid search on two hyper-parameters that exactly determines the memory bank, *i.e.*, the starting epoch for recording and the number of epochs to be recorded. Specifically, the starting epoch signifies the training epoch at which we start to record predictions to construct the memory bank; the number of epochs indicates the total number of training epochs during which predictions are recorded. The grid search is conducted on node classification task and adopt GCN as the base model in Algorithm 2.

Fig 5 summarize the results of grid search in terms of classification accuracy on three citation graphs. We can observe that if the starting epoch is too early or the number of epochs is not sufficient, performance may be suboptimal. It is because, during the early phases of training, the model changes dramatically, potentially introducing errors when computing MoDis. A limited number of epochs fails to fully capture training dynamics. Therefore, a simple yet effective strategy to determine the hyper-parameters is to enlarge the memory bank by increasing the number of epochs, which however increases computational overhead, reducing efficiency.

## 5 RELATED WORK AND DISCUSSION

### 5.1 Comparisons to Existing Methods

Pseudo-labeling [20] was introduced as an effective and widely adopted strategy in semi-supervised learning, which predicted labels of unlabeled data, as a data augmentation of limited labeled data. Considering extensive unlabeled nodes in a single graph, especially for transductive graph learning task, [21] and [35] extended pseudo-labeling algorithms to the field of SSGL. The wildly-used uncertainty measure for pseudo-labeling is confidence score, and a comprehensive overview of uncertainty is given in the survey [12]. Based on confidence score, recent works attempt to improve the quality of pseudo labels by using topological information in[42] and [34], and by refining training framework in [22] and [44].

**The principal novelty of our proposed** MoDis**, distinct from existing methods, is the perspective of training dynamics,**

which introduces a novel way for estimating prediction uncertainty. In essence, the most significant contribution of this paper lies in revealing that training dynamics contain valuable information regarding prediction uncertainty in graph learning. In contrast, the confidence score is calculated from a softmax probability distribution provided by the model after training; Monte-Carlo dropout [10], another popular method for measuring uncertainty, enables dropout during testing and performs multiple forward passes through the network after training. The proposed MoDis is orthogonal to the existing uncertainty measures.

According to taxonomy in recent surveys on uncertainty in deep learning [1, 12], the proposed MoDis can be viewed as a special ensemble method, where models at different training epochs are treated as ensemble members. The ensemble method derives a prediction based on the aggregated predictions from multiple ensemble members (models), whose assumption is that a group of decision-makers typically makes better decisions than an individual. In addition, ensemble methods provide a way to estimate prediction uncertainty by evaluating the variety among the member's predictions [19]. A similar self-ensembling strategy is employed to filter out noise labels [25]. Ensemble-based uncertainty estimation requires training multiple models as members, while the calculation of MoDis is only based on a single training process. Thus, one important advantage of the proposed MoDis is computational efficiency, compared to classical ensemble-based methods.

### 5.2 Algorithm Limitation — No Free Lunch

The proposed algorithm also follows the "no free lunch" theorem [41]. Although MoDis based pseudo-labeling algorithm has time complexity comparable to popular confidence score-based methods, it introduces additional space complexity due to the usage of the memory bank $\mathcal{M}$ to capture training dynamics. The space complexity of $\mathcal{M}$ is $O(n|\mathcal{T}|)$, where $n$ denotes the number of nodes in graph and $|\mathcal{T}|$ represents the number of sampled epochs to construct $\mathcal{M}$. In practice, $n$ substantially exceeds $|\mathcal{T}|$, rendering the additional complexity to be $O(|n|)$, linearly with the size of graph.

## 6 CONCLUSION

In this study, we uncover that training dynamics offer significant insights into prediction uncertainty in graph learning. Leveraging training dynamics, we proposed a novel uncertainty measure, memory disagreement (MoDis) for pseudo-labeling, which is able to identify both correct and informative pseudo-labeled nodes in SSGL. This presents an alternative way to avoid the limitations of the widely-used confidence score. The implementation of MoDis showcases its adaptability, as evidenced by its successful application across various graph neural networks and its superior performance on multiple benchmark graph datasets. The enhanced correctness and information gain of pseudo labels offered by MoDis herald a promising future for its widespread adoption. This paper sets a new benchmark in pseudo-labeling based SSGL.

Our future work includes three parts: (i) Developing the theoretical foundations of MoDis, especially a guidance for constructing an effective memory bank; (ii) Designing MoDis uncertainty aware graph neural networks to deal with noise in graph data; (iii) Expanding the application of MoDis from graph learning to other domains of semi-supervised learning, such as computer vision.

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

# APPENDIX

This appendix consists of two sections. We describe experimental details in the first section, and analyze number of pseudo-labeled nodes required for semi-supervised learning in the second section.

## A EXPERIMENTAL DETAILS

The following four subsections present details of the four corresponding experiments in the main paper, respectively. The details mainly include experiment protocols and hyper-parameter settings.

### A.1 Details of Experiment 1

Experiment 1 is to evaluate the quality of pseudo labels given by the proposed MoDis. We evaluate the quality from two beforementioned aspects, correctness and information gain. We adopt GCN as base model with the same hyper-parameters in [18]. Datasets includ Cora, Citeseer, Pubmed, CoraFull, AmazonCS, and AmazonPhoto. We only assign 3 labeled nodes per class in training, $i.e.$, $L/C = 3$.

**Correctness evaluation.** In this experiment, we first predict labels for all unlabeled nodes using a pre-trained GCN model. We then rank these predicted labels according to the three uncertainty measures in descending order, $i.e.$, MoDis, confidence score, and AUM, respectively. Each ranked label sequence is further segmented into 10 equal groups. Intuitively, predicted labels in top groups should be prioritized as pseudo labels because they have higher correctness than the rest. We subsequently calculate correctness of the top 5 groups of each sequences in terms of $error\ ratio$ that is defined in the main paper. Based on the calculated correctness, we compare the quality of pseudo labels given by the three measures, as shown in Fig. 4 in the main paper.

**Information gain evaluation.** We employ model perturbation $\rho$ to assess the information gain introduced by the pseudo-labeled nodes. This metric measures gradient changes resulting from the inclusion of pseudo-labeled nodes, and the detailed calculation of $\rho$ is presented in the main paper.

In this experiment, we first train a GCN model until it reaches convergence. Then, we identify and select the a pre-given number, CPL/C, of correct pseudo labels by using the three uncertainty measures, $i.e.$, MoDis, confidence score, and AUM, respectively. We subsequently retrain the GCN model using these pseudo labels and calculate the model perturbation $\rho$. It's important to note that for a meaningful evaluation of information gain, only correct pseudo labels are utilized. The model perturbations corresponding to the three measures are summarized in Table 2 in the main paper.

### A.2 Details of Experiment 2

In experiment 2, node classification is conducted on eight benchmark graph datasets. To thoroughly evaluate our proposed method, we establish scenarios with various label information in data. We adjust the number of labeled nodes per category, denoted as $L/C$, to values of 3, 5, 10, and 20 in all datasets. When $L/C = 20$, we follow the original partition provided by the datasets as referenced in the main paper. For other $L/C$ setting, we conduct 10 random splits. Each of these splits involves randomly selecting a subset of nodes from the training set where $L/C = 20$.

Utilizing the curated data, we employ Algorithm 1 to generate pseudo labels and use Algorithm 2 for node label prediction. For

baseline comparisons, we substitute MoDis MoDis with either the confidence score or AUM within the respective algorithm.

Utilizing the prepared data, we employ Algorithm 1 to generate pseudo labels and use Algorithm 2 for predicting node labels. For baseline comparisons, we substitute MoDis with either the confidence score or AUM in the corresponding algorithm. In the algorithms, we adopt four widely used graph neural network models as the base model, namely GCN [18], GAT [37], APPNP[11] and GCNII [7], following the same hyper-parameter settings in their original papers. For all methods and settings, we run the experiment 10 times and report the average accuracy, are summarized in Table 3 and 4 in the main paper.

We employ grid search to determine the optimal hyper-parameters for both our method and baseline methods. The specific hyper-parameter configurations can be found in Table 6, 7, 8, and 9. In these tables:

- "Hidden num" denotes the number of neurons in the network's hidden layer;
- "No. of layers" indicates the total number of layers within the network;
- "Learning rate" specifies the learning rate utilized;
- "Weight decay" denotes the weight decay associated with the L2 regularization term;
- "K" indicates the number of stage in the self-training algorithm;
- "P" denotes the total number of pseudo-labeled nodes used in each category;
- "Starting epoch" signifies the training epoch at which we start to record predictions to construct the memory bank;
- "Number of epochs" indicates the total number of training epochs during which predictions are recorded by the memory bank;
- "$\gamma$" represent the temperature coefficient in the Sharpen function.

### A.3 Details of Experiment 3

Experiment 3 aims to validate our proposed method on OOD (Out-of-Distribution) nodes in a graph. In this graph learning scenario with OOD nodes, some unlabeled nodes originate from distributions distinct from the labeled nodes. For clarity, we categorize unlabeled nodes that share the same distribution with labeled nodes as ID (in-distribution) nodes, commonly known as inliers. While, unlabeled nodes from different distributions are identified as OOD (out-of-distribution) nodes, or outliers. There are two tasks in this OOD experiment: (1) semi-supervised node classification for the ID nodes, and (2) OOD nodes detection. We use classification accuracy (ACC) to evaluate the semi-supervised node classification task, and employ false positive rate at 95% true positive rate (FPR@95) to evaluate the OOD nodes detection task.

We leverage OODGAT, a graph attention network specially designed for OOD data, as base model in our pseudo labeling algorithms, following the same hyper-parameter setting specified in OODGAT's original paper [32]. The experiment is conducted on four graph datasets, Cora, AmazonComputers, AmazonPhoto, and CoauthorCS. For every dataset, nodes are partitioned into ID classes

and OOD classes, with the specific configurations detailed in Table 10. We form the training set by randomly choosing three nodes from each ID class; the validation set is composed of ten randomly selected nodes from ID classes. All remaining nodes constitute the test set for the OOD node detection task; the remaining nodes in ID classes for the semi-supervised node classification task.

We train model on training set that only contains ID nodes. When selecting pseudo-labeled nodes, we consider all unlabelled nodes that are in ID or OOD classes. It's evident that taking OOD nodes as pseudo-labeled nodes can introduce large misleading information to the model. For node classification, we follow the same experiment protocols as that in experiment 2. For OOD detection, we use the attention scores generated by OODGAT to identify whether a unlabelled node belongs to ID or OOD classes, in an unsupervised manner. The hyper-parameters used in this experiment are summarized in Table 10. In this tables:

- "Hidden num" denotes the number of neurons in the network's hidden layer;
- "No. of layers" indicates the total number of layers within the network;
- "Learning rate" specifies the learning rate utilized;
- "Weight decay" denotes the weight decay associated with the L2 regularization term;
- "K" indicates the number of stage in the self-training algorithm;
- "P" denotes the total number of pseudo-labeled nodes used in each category;
- "Starting epoch" signifies the training epoch at which we start to record predictions to construct the memory bank;
- "Number of epochs" indicates the total number of training epochs during which predictions are recorded by the memory bank;
- "$\gamma$" represent the temperature coefficient in the Sharpen function.

## A.4 Details of Experiment 4

The memory bank serves as the foundation of our proposed method, facilitating the capture of training dynamics. Two key hyper-parameters govern the memory bank: the starting epoch and the number of epochs. The starting epoch signifies the training epoch at which we start to record predictions to construct the memory bank; the number of epochs indicates the total number of training epochs during which predictions are recorded by the memory bank.

To analyze the influence of these hyper-parameters on the algorithm performance, we conduct a grid search on node classification on three citation graphs, Cora, Citeseer, and Pubmed. For the starting epoch, the explored values span [0, 10, 20, 50, 100], and for the number of epochs, the range includes [1, 10, 20, 50, 100, 150, 200, 300]. We adopt the GCN as base model in our algorithms, and training set contains only three labeled node per catergory, $L/C = 3$. We follow the same experiment protocols as that in experiment 2. The outcomes of grid search are visualized via a heatmap in Fig. 5 in the main paper.

## B ANALYSIS OF NUMBER OF PSEUDO-LABELED NODES REQUIRED

In the node classification experiments, we take a grid search for the optimal hyper-parameters in all datasets. Notably, there was a marked variance in the total number of pseudo-labeled nodes (denoted by parameter P in Table 6, 7, 8, and 9), comparing across different datasets.

We observe a correlation between the number of pseudo labels required and the average degree of a graph. This relationship is visualized using a scatter diagram, as illustrated in Fig. 6. In Fig. 6, each blue point represents the characteristics of a specific dataset: the x-axis indicates the average degree of the graphs, while the y-axis signifies the optimal number of pseudo-labeled nodes provided by the grid search. Notably, the y-axis is on a logarithmic scale. This observation can be reasonably explained by noting that as the average degree of the graph rises, the supervised information from both labeled and pseudo-labeled nodes propagates more rapidly through the GCN model. Consequently, there's a reduced need to add additional pseudo-labeled nodes to effectively spread the valuable supervised information throughout the graph.

We further validate this hypothesis by taking a Kendall correlation test between average degree of graph and the parameter P for $L/C = 3$ scenarios in Table 6, 7, 8, and 9. The resulting Kendall correlation coefficient is -0.67 with an associated p-value of 0.02. This outcome suggests a strong negative correlation between the parameter P and the graph's average node degree. We will deeply explore this interesting correlation in our future work.

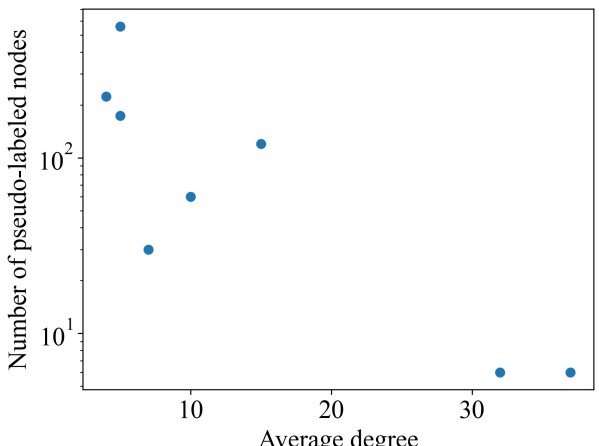

**Figure 6: The scatter diagram to visualize a negative correlation between the number of pseudo labels required and the average degree of a graph. Each blue point denotes the characteristics of a dataset. The x-axis denotes average degree of graphs, while the y-axis denotes number of pseudo-labeled nodes required. The y-axis is on a logarithmic scale.**

Received 20 February 2007; revised 12 March 2009; accepted 5 June 2009

**Table 6: The hyper-parameters for GCN with the pseudo-labeling algorithm**

| Dataset | Cora | | | | Citeseer | | | | Pubmed | | | | CoraFull | | | |
|---|---|---|---|---|---|---|---|---|---|---|---|---|---|---|---|---|
| L/C | 3 | 5 | 10 | 20 | 3 | 5 | 10 | 20 | 3 | 5 | 10 | 20 | 3 | 5 | 10 | 20 |
| Hidden Num | 16 | 16 | 16 | 16 | 16 | 16 | 16 | 16 | 16 | 16 | 16 | 16 | 64 | 64 | 64 | 64 |
| No. of layers | 2 | 2 | 2 | 2 | 2 | 2 | 2 | 2 | 2 | 2 | 2 | 2 | 2 | 2 | 2 | 2 |
| Learning Rate | 5E-2 | 5E-2 | 5E-2 | 5E-2 | 5E-2 | 5E-2 | 5E-2 | 5E-2 | 5E-2 | 5E-2 | 5E-2 | 5E-2 | 5E-2 | 5E-2 | 5E-2 | 5E-2 |
| Weight Decay | 5E-4 | 5E-4 | 5E-4 | 5E-4 | 5E-4 | 5E-4 | 5E-4 | 5E-4 | 5E-4 | 5E-4 | 5E-4 | 5E-4 | 5E-4 | 5E-4 | 5E-4 | 5E-4 |
| K | 2 | 2 | 2 | 2 | 7 | 7 | 3 | 5 | 5 | 5 | 2 | 3 | 3 | 3 | 3 | 3 |
| Starting Epoch | 0 | 20 | 50 | 50 | 50 | 50 | 50 | 20 | 50 | 50 | 50 | 50 | 100 | 100 | 100 | 100 |
| Number of Epochs | 150 | 100 | 100 | 100 | 150 | 100 | 100 | 50 | 100 | 100 | 100 | 100 | 100 | 100 | 100 | 100 |
| P | 174 | 192 | 112 | 112 | 223 | 214 | 90 | 61 | 560 | 700 | 677 | 480 | 30 | 12 | 15 | 6 |
| $\gamma$ | 0.1 | 0.1 | 0.01 | 0.01 | 0.9 | 0.6 | 0.1 | 0.4 | 0.01 | 0.01 | 0.01 | 0.01 | 0.1 | 0.1 | 0.1 | 0.1 |
| Dataset | AmazonCS | | | | AmazonPhoto | | | | CoauthorCS | | | | CoauthorPhy | | | |
| L/C | 3 | 5 | 10 | 20 | 3 | 5 | 10 | 20 | 3 | 5 | 10 | 20 | 3 | 5 | 10 | 20 |
| Hidden Num | 64 | 64 | 64 | 64 | 64 | 64 | 64 | 64 | 64 | 64 | 64 | 64 | 64 | 64 | 64 | 64 |
| No. of layers | 2 | 2 | 2 | 2 | 2 | 2 | 2 | 2 | 2 | 2 | 2 | 2 | 2 | 2 | 2 | 2 |
| Learning Rate | 5E-3 | 5E-3 | 5E-3 | 5E-3 | 5E-3 | 5E-3 | 5E-3 | 5E-3 | 5E-3 | 5E-3 | 5E-3 | 5E-3 | 5E-3 | 5E-3 | 5E-3 | 5E-3 |
| Weight Decay | 5E-3 | 5E-3 | 5E-3 | 5E-3 | 5E-3 | 5E-3 | 5E-3 | 5E-3 | 5E-3 | 5E-3 | 5E-3 | 5E-3 | 5E-3 | 5E-3 | 5E-3 | 5E-3 |
| K | 3 | 3 | 3 | 3 | 3 | 3 | 3 | 3 | 3 | 3 | 3 | 3 | 3 | 3 | 3 | 3 |
| Starting Epoch | 100 | 100 | 100 | 100 | 100 | 100 | 100 | 100 | 100 | 100 | 100 | 100 | 100 | 100 | 100 | 100 |
| Number of Epochs | 100 | 100 | 100 | 100 | 100 | 100 | 100 | 100 | 100 | 100 | 100 | 100 | 100 | 100 | 100 | 100 |
| P | 6 | 6 | 6 | 6 | 6 | 6 | 6 | 18 | 60 | 60 | 60 | 60 | 120 | 120 | 90 | 90 |
| $\gamma$ | 0.1 | 0.1 | 0.1 | 0.1 | 0.1 | 0.1 | 0.1 | 0.1 | 0.1 | 0.1 | 0.1 | 0.1 | 0.1 | 0.1 | 0.1 | 0.1 |

**Table 7: The hyper-parameters for GAT with the pseudo-labeling algorithm**

| Dataset | Cora | | | | Citeseer | | | | Pubmed | | | | CoraFull | | | |
|---|---|---|---|---|---|---|---|---|---|---|---|---|---|---|---|---|
| L/C | 3 | 5 | 10 | 20 | 3 | 5 | 10 | 20 | 3 | 5 | 10 | 20 | 3 | 5 | 10 | 20 |
| Hidden Num | 8 | 8 | 8 | 8 | 8 | 8 | 8 | 8 | 8 | 8 | 8 | 8 | 32 | 32 | 32 | 32 |
| No. of layers | 2 | 2 | 2 | 2 | 2 | 2 | 2 | 2 | 2 | 2 | 2 | 2 | 2 | 2 | 2 | 2 |
| Learning Rate | 5E-3 | 5E-3 | 5E-3 | 5E-3 | 5E-3 | 5E-3 | 5E-3 | 5E-3 | 5E-3 | 5E-3 | 5E-3 | 5E-3 | 5E-2 | 5E-2 | 5E-2 | 5E-2 |
| Weight Decay | 5E-4 | 5E-4 | 5E-4 | 5E-4 | 5E-4 | 5E-4 | 5E-4 | 5E-4 | 5E-4 | 5E-4 | 5E-4 | 5E-4 | 5E-4 | 5E-4 | 5E-4 | 5E-4 |
| K | 3 | 4 | 4 | 3 | 6 | 3 | 2 | 2 | 4 | 4 | 3 | 2 | 3 | 3 | 3 | 3 |
| Starting Epoch | 20 | 100 | 10 | 50 | 50 | 50 | 50 | 50 | 50 | 50 | 50 | 20 | 100 | 100 | 100 | 100 |
| Number of Epochs | 100 | 150 | 150 | 100 | 100 | 100 | 100 | 100 | 100 | 100 | 100 | 100 | 100 | 100 | 100 | 100 |
| P | 96 | 150 | 48 | 42 | 212 | 160 | 84 | 60 | 480 | 600 | 320 | 36 | 30 | 12 | 15 | 6 |
| $\gamma$ | 0.5 | 0.5 | 0.5 | 0.5 | 0.1 | 0.1 | 0.1 | 0.1 | 0.01 | 0.01 | 0.01 | 0.01 | 0.1 | 0.1 | 0.1 | 0.1 |
| Dataset | AmazonCS | | | | AmazonPhoto | | | | CoauthorCS | | | | CoauthorPhy | | | |
| L/C | 3 | 5 | 10 | 20 | 3 | 5 | 10 | 20 | 3 | 5 | 10 | 20 | 3 | 5 | 10 | 20 |
| Hidden Num | 32 | 32 | 32 | 32 | 32 | 32 | 32 | 32 | 32 | 32 | 32 | 32 | 32 | 32 | 32 | 32 |
| No. of layers | 2 | 2 | 2 | 2 | 2 | 2 | 2 | 2 | 2 | 2 | 2 | 2 | 2 | 2 | 2 | 2 |
| Learning Rate | 5E-3 | 5E-3 | 5E-3 | 5E-3 | 5E-3 | 5E-3 | 5E-3 | 5E-3 | 5E-3 | 5E-3 | 5E-3 | 5E-3 | 5E-3 | 5E-3 | 5E-3 | 5E-3 |
| Weight Decay | 5E-3 | 5E-3 | 5E-3 | 5E-3 | 5E-3 | 5E-3 | 5E-3 | 5E-3 | 5E-3 | 5E-3 | 5E-3 | 5E-3 | 5E-3 | 5E-3 | 5E-3 | 5E-3 |
| K | 3 | 3 | 3 | 3 | 3 | 3 | 3 | 3 | 3 | 3 | 3 | 3 | 3 | 3 | 3 | 3 |
| Starting Epoch | 100 | 100 | 100 | 150 | 150 | 150 | 150 | 100 | 100 | 100 | 100 | 100 | 100 | 100 | 100 | 100 |
| Number of Epochs | 100 | 100 | 100 | 50 | 50 | 50 | 50 | 100 | 100 | 100 | 100 | 100 | 100 | 100 | 100 | 100 |
| P | 6 | 6 | 6 | 3 | 3 | 12 | 3 | 12 | 60 | 100 | 100 | 100 | 100 | 100 | 100 | 100 |
| $\gamma$ | 0.01 | 0.01 | 0.01 | 0.1 | 0.1 | 0.1 | 0.1 | 0.1 | 0.1 | 0.5 | 0.5 | 0.5 | 0.5 | 0.5 | 0.5 | 0.5 |

**Table 8: The hyper-parameters for APPNP with the pseudo-labeling algorithm**

| Dataset | Cora | | | | Citeseer | | | | Pubmed | | | | CoraFull | | | |
|---|---|---|---|---|---|---|---|---|---|---|---|---|---|---|---|---|
| L/C | 3 | 5 | 10 | 20 | 3 | 5 | 10 | 20 | 3 | 5 | 10 | 20 | 3 | 5 | 10 | 20 |
| Hidden Num | 64 | 64 | 64 | 64 | 64 | 64 | 64 | 64 | 64 | 64 | 64 | 64 | 64 | 64 | 64 | 64 |
| No. of layers | 2 | 2 | 2 | 2 | 2 | 2 | 2 | 2 | 2 | 2 | 2 | 2 | 2 | 2 | 2 | 2 |
| Learning Rate | 1E-2 | 1E-2 | 1E-2 | 1E-2 | 1E-2 | 1E-2 | 1E-2 | 1E-2 | 1E-2 | 1E-2 | 1E-2 | 1E-2 | 5E-2 | 5E-2 | 5E-2 | 5E-2 |
| Weight Decay | 5E-4 | 5E-4 | 5E-4 | 5E-4 | 5E-4 | 5E-4 | 5E-4 | 5E-4 | 5E-4 | 5E-4 | 5E-4 | 5E-4 | 5E-4 | 5E-4 | 5E-4 | 5E-4 |
| K | 5 | 4 | 3 | 3 | 2 | 4 | 3 | 3 | 3 | 3 | 2 | 4 | 3 | 3 | 3 | 3 |
| Starting Epoch | 100 | 100 | 100 | 100 | 100 | 100 | 100 | 100 | 100 | 100 | 100 | 100 | 100 | 100 | 100 | 100 |
| Number of Epochs | 100 | 100 | 100 | 200 | 50 | 50 | 50 | 50 | 100 | 100 | 50 | 100 | 100 | 100 | 100 | 100 |
| P | 111 | 178 | 63 | 63 | 111 | 96 | 96 | 54 | 675 | 720 | 708 | 328 | 30 | 12 | 15 | 6 |
| $\gamma$ | 0.5 | 0.5 | 0.5 | 0.5 | 0.1 | 0.1 | 0.1 | 0.1 | 0.01 | 0.01 | 0.01 | 0.01 | 0.1 | 0.1 | 0.1 | 0.1 |
| Dataset | AmazonCS | | | | AmazonPhoto | | | | CoauthorCS | | | | CoauthorPhy | | | |
| L/C | 3 | 5 | 10 | 20 | 3 | 5 | 10 | 20 | 3 | 5 | 10 | 20 | 3 | 5 | 10 | 20 |
| Hidden Num | 64 | 64 | 64 | 64 | 64 | 64 | 64 | 64 | 64 | 64 | 64 | 64 | 64 | 64 | 64 | 64 |
| No. of layers | 2 | 2 | 2 | 2 | 2 | 2 | 2 | 2 | 2 | 2 | 2 | 2 | 2 | 2 | 2 | 2 |
| Learning Rate | 5E-3 | 5E-3 | 5E-3 | 5E-3 | 5E-3 | 5E-3 | 5E-3 | 5E-3 | 5E-3 | 5E-3 | 5E-3 | 5E-3 | 5E-3 | 5E-3 | 5E-3 | 5E-3 |
| Weight Decay | 5E-3 | 5E-3 | 5E-3 | 5E-3 | 5E-3 | 5E-3 | 5E-3 | 5E-3 | 5E-3 | 5E-3 | 5E-3 | 5E-3 | 5E-3 | 5E-3 | 5E-3 | 5E-3 |
| K | 3 | 3 | 3 | 3 | 3 | 3 | 3 | 3 | 3 | 3 | 3 | 3 | 3 | 3 | 3 | 3 |
| Starting Epoch | 100 | 100 | 100 | 150 | 100 | 100 | 100 | 100 | 100 | 100 | 100 | 100 | 100 | 100 | 100 | 100 |
| Number of Epochs | 100 | 100 | 100 | 50 | 100 | 100 | 100 | 100 | 100 | 100 | 100 | 100 | 100 | 100 | 100 | 100 |
| P | 6 | 6 | 3 | 6 | 3 | 6 | 6 | 9 | 60 | 60 | 60 | 60 | 96 | 120 | 96 | 90 |
| $\gamma$ | 0.1 | 0.1 | 0.01 | 0.01 | 0.4 | 0.4 | 0.4 | 0.4 | 0.01 | 0.1 | 0.1 | 0.1 | 0.01 | 0.1 | 0.1 | 0.1 |

**Table 9: The hyper-parameters for GCNII with the pseudo-labeling algorithm**

| Dataset | Cora | | | | Citeseer | | | | Pubmed | | | | CoraFull | | | |
|---|---|---|---|---|---|---|---|---|---|---|---|---|---|---|---|---|
| L/C | 3 | 5 | 10 | 20 | 3 | 5 | 10 | 20 | 3 | 5 | 10 | 20 | 3 | 5 | 10 | 20 |
| Hidden Num | 64 | 64 | 64 | 64 | 256 | 256 | 256 | 256 | 256 | 256 | 256 | 256 | 256 | 256 | 256 | 256 |
| No. of layers | 64 | 64 | 64 | 64 | 32 | 32 | 32 | 32 | 16 | 16 | 16 | 16 | 32 | 32 | 32 | 32 |
| Learning Rate | 1E-2 | 1E-2 | 1E-2 | 1E-2 | 1E-2 | 1E-2 | 1E-2 | 1E-2 | 1E-2 | 1E-2 | 1E-2 | 1E-2 | 5E-2 | 5E-2 | 5E-2 | 5E-2 |
| Weight Decay1 | 1E-2 | 1E-2 | 1E-2 | 1E-2 | 1E-2 | 1E-2 | 1E-2 | 1E-2 | 1E-2 | 1E-2 | 1E-2 | 1E-2 | 5E-2 | 5E-2 | 5E-2 | 5E-2 |
| Weight Decay2 | 5E-4 | 5E-4 | 5E-4 | 5E-4 | 5E-4 | 5E-4 | 5E-4 | 5E-4 | 5E-4 | 5E-4 | 5E-4 | 5E-4 | 5E-4 | 5E-4 | 5E-4 | 5E-4 |
| K | 5 | 4 | 3 | 3 | 2 | 4 | 3 | 3 | 3 | 3 | 2 | 4 | 3 | 3 | 3 | 3 |
| Starting Epoch | 400 | 400 | 400 | 400 | 400 | 400 | 400 | 400 | 400 | 400 | 400 | 400 | 400 | 400 | 400 | 400 |
| Number of Epochs | 100 | 100 | 100 | 100 | 100 | 100 | 100 | 100 | 100 | 100 | 100 | 100 | 100 | 100 | 100 | 100 |
| P | 111 | 178 | 105 | 63 | 111 | 96 | 96 | 54 | 675 | 720 | 708 | 328 | 30 | 15 | 15 | 6 |
| $\gamma$ | 0.5 | 0.5 | 0.5 | 0.5 | 0.1 | 0.1 | 0.1 | 0.1 | 0.01 | 0.01 | 0.01 | 0.01 | 0.1 | 0.1 | 0.1 | 0.1 |
| Dataset | AmazonCS | | | | AmazonPhoto | | | | CoauthorCS | | | | CoauthorPhy | | | |
| L/C | 3 | 5 | 10 | 20 | 3 | 5 | 10 | 20 | 3 | 5 | 10 | 20 | 3 | 5 | 10 | 20 |
| Hidden Num | 64 | 64 | 64 | 64 | 64 | 64 | 64 | 64 | 64 | 64 | 64 | 64 | 64 | 64 | 64 | 64 |
| No. of layers | 64 | 64 | 64 | 64 | 64 | 64 | 64 | 64 | 64 | 64 | 64 | 64 | 64 | 64 | 64 | 64 |
| Learning Rate | 1E-2 | 1E-2 | 1E-2 | 1E-2 | 1E-2 | 1E-2 | 1E-2 | 1E-2 | 1E-2 | 1E-2 | 1E-2 | 1E-2 | 1E-2 | 1E-2 | 1E-2 | 1E-2 |
| Weight Decay1 | 1E-2 | 1E-2 | 1E-2 | 1E-2 | 1E-2 | 1E-2 | 1E-2 | 1E-2 | 1E-2 | 1E-2 | 1E-2 | 1E-2 | 1E-2 | 1E-2 | 1E-2 | 1E-2 |
| Weight Decay2 | 5E-4 | 5E-4 | 5E-4 | 5E-4 | 5E-4 | 5E-4 | 5E-4 | 5E-4 | 5E-4 | 5E-4 | 5E-4 | 5E-4 | 5E-4 | 5E-4 | 5E-4 | 5E-4 |
| K | 3 | 3 | 3 | 3 | 3 | 3 | 3 | 3 | 3 | 3 | 3 | 3 | 3 | 3 | 3 | 3 |
| Starting Epoch | 400 | 400 | 400 | 400 | 150 | 150 | 150 | 400 | 400 | 400 | 400 | 400 | 400 | 400 | 400 | 400 |
| Number of Epochs | 100 | 100 | 100 | 100 | 350 | 350 | 350 | 100 | 100 | 100 | 100 | 100 | 100 | 100 | 100 | 100 |
| P | 6 | 6 | 6 | 6 | 3 | 3 | 3 | 12 | 60 | 60 | 60 | 60 | 120 | 120 | 90 | 90 |
| $\gamma$ | 0.01 | 0.01 | 0.01 | 0.1 | 0.1 | 0.1 | 0.1 | 0.1 | 0.1 | 0.1 | 0.1 | 0.1 | 0.1 | 0.1 | 0.1 | 0.1 |

Table 10: The hyper-parameters for OODGAT with the pseudo-labeling algorithm

| Dataset | Cora | AmazonCS | AmazonPhoto | CoauthorCS |
|---|---|---|---|---|
| ID Classes | [4,2,5,6] | [8,1,2,7,6] | [3,4,5,2,0] | [5,11,10,7,14,8,12,6] |
| Splits | [3,10,1000] | [3,10,5000] | [3,10,3000] | [3,10,8000] |
| Continuous | False | False | False | False |
| Weight Consistent | 2 | 2 | 3 | 4 |
| Weight Entropy | 0.05 | 0.05 | 0.10 | 0.05 |
| Weight Discrepancy | 0.005 | 0.005 | 0.005 | 0.005 |
| Margin | 0.6 | 0.4 | 0.4 | 0.6 |
| Heads | 4 | 4 | 4 | 4 |
| K | 3 | 3 | 3 | 3 |
| P | 12 | 4 | 6 | 6 |
| Starting Epoch | 20 | 50 | 100 | 30 |
| Number of Epochs | 250 | 500 | 600 | 250 |
| $\gamma$ | 0.3 | 0.1 | 0.01 | 0.3 |

