# OpenReview forum: "Memory Disagreement: A Pseudo-Labeling Measure from Training Dynamics for Semi-supervised Graph Learning"
_ACM.org/TheWebConf/2024/Conference — TheWebConf24_

### Official Review · Reviewer_9xtA · 2023-10-30

**Novelty:** 4
**Technical Quality:** 5

**Review:**

In the paradigm of SSGL, pseudo-labeling is widely adopted to overcome the sparsity of labeled nodes. However, the confidence score, one of the most frequent methods for generating pseudo labels, suffers from poor calibration and out-of-distribution data. In this paper, we utilize training dynamics to define a new metric, named MoDis, to evaluate the degree of disagreement among model predictions at different training epochs, thereby identifying pseudo-labeled nodes.

Authors conduct two case studies to evaluate the rationale of MoDis, and comprehensive evaluations to validate the effectiveness.

**Questions:**

Weaknesses
- This paper seems like a lack of novelty. Because, in fact, MoDis is applied to evaluate the consistency degree of predictions of unlabeled nodes. If a node is always classified into a certain class in many consecutive epochs, it is hard to say that assigning pseudo labels to these nodes will have an important impact.
- This paper lacks theoretical analysis to prove the effectiveness of MoDis.
- There are many other methods to generate pseudo labels [1,2,3]. Authors should evaluate the performance of MoDis with these methods.
- In Table 3 and 4, we can observe when we adopt Confidence and AUM to generate pseudo labels and conduct a multi-stage pseudo labelling, the node classification accuracy decreases instead. Authors should make some clarifications on this phenomenon.
- Authors should conduct experiments to evaluate the impact of the size of memory bank on the performance of MoDis.

[1] Lee, D.H. Pseudo-Label : The Simple and Efficient Semi-Supervised Learning Method for Deep Neural Networks. ICML 2013.
[2] Pham, H., et al. Meta Pseudo Labels. CVPR 2021.
[3] Hu, Z., et al. SimPLE: Similar Pseudo Label Exploitation for Semi-Supervised Classification. CVPR2021.

**Reviewer Confidence:**

4: The reviewer is certain that the evaluation is correct and very familiar with the relevant literature

**Scope:**

3: The work is somewhat relevant to the Web and to the track, and is of narrow interest to a sub-community

---

### Official Review · Reviewer_Fz19 · 2023-11-05

**Novelty:** 2
**Technical Quality:** 5

**Review:**

This paper suggest MoDis, a thresholding measure to select reliable pseudo-label from model prediction. MoDis is entropy of softmax trajectory for a node in a graph and measures consistency in predicted class. In experiments using a number of dataset, MoDis outperforms other measures in semi-supervised node classification task and selects more reliable pseudo-labels.

Pros
1. Simple and effective approach to ensure reliability of pseudo-labels
2. Diverse analysis on the quality of MoDis
3. Robust experiements to verify effectiveness

Cons
1. Low novelty due to similar work. Please see questions.
2. Lack of hyperparameter sensitivity experiment

**Questions:**

1. Entropy of softmax trajectory is well-known method, such as SELFIE [1]. In SELFIE, the entropy is used to select reliable labels from classifier trained by noisy labels. It seems that the method is too similar and the only difference is the application that the entropy is used for graph node classification. Please inform me if there is a significant difference between MoDis and SELFIE.

2. MoDis is a metric that can be applied to any instance and seems have no strong relationship to graph dataset. Can you elaborate more on the reason why MoDis should be used for node classification?

3. Hyperparameter sensitivity on the threshold $\tau$ and the size of memory bank should exist to show that MoDis is robust to hyperparameters.

[1] Song, H., Kim, M. Lee, J.. (2019). SELFIE: Refurbishing Unclean Samples for Robust Deep Learning. ICML.

**Ethics Review Description:**

No issue

**Reviewer Confidence:**

3: The reviewer is confident but not certain that the evaluation is correct

**Scope:**

3: The work is somewhat relevant to the Web and to the track, and is of narrow interest to a sub-community

---

### Official Review · Reviewer_YGwQ · 2023-11-19

**Novelty:** 6
**Technical Quality:** 5

**Review:**

## Summary

The authors propose MoDis, a novel uncertainty measure for pseudo-labeling for semi-supervised graph learning. The main novelty is over existing methods such as confidence score is that MoDis uses the entropy in label prediction along the training process of each node as the uncertainty measure. The extensive experiment shows the strong empirical performance of the method.

## Strengths

(+) Clever and novel idea. It is simple (in a good way) yet effective.

(+) The paper is well written and the key idea is conveyed very clearly.

(+) The experiment is extensive and shows a strong advantage of the proposed method.

(+) The authors discuss the limitations of their method, such as additional space complexity which I think is super great. This kind of honest discussion is really rare in nowadays research and should definitely be encouraged.

## Weaknesses

To be honest, the weakness below is minor to me. Still, it can make the paper even better.

(-) All the results are reported without error bars. I hope the authors can include it so that readers can better judge whether the performance gain is statistically significant or not.


## Detail comments

I really like the paper. The authors did a great job of conveying the key idea and why it is potentially better than existing approaches. They also conducted several case studies (i.e., Figures 2 and 3) to validate their idea empirically.

Although I think the paper is already good in its current shape, I find it can be even better if the weakness above can be addressed. Also, I list several questions below simply out of my curiosity. Perhaps it can be a good future direction.

**Questions:**

1.	Currently the authors simply use the prediction of the entire training trajectory. I wonder if setting some importance weight can lead to better result? We know that at the beginning of the training phase, the prediction can be quite random. Not sure if such an importance weight is helpful or harmful.

2.	Can we also leverage the “confidence score” along the training trajectory for a better version of MoDis?

It is totally fine if the authors cannot answer these questions concretely for now. Yet I would appreciate any thoughts from the authors.

**Reviewer Confidence:**

3: The reviewer is confident but not certain that the evaluation is correct

**Scope:**

3: The work is somewhat relevant to the Web and to the track, and is of narrow interest to a sub-community

---

### Official Review · Reviewer_AUiV · 2023-11-25

**Novelty:** 4
**Technical Quality:** 5

**Review:**

This paper presents a simple quantity, called memory disagreement (MoDis), for measuring node uncertainty for semi-supervised classification of nodes using graph neural networks.
It is based on training dynamics to calculate the entropy of a node. Some experimental results are demonstrated to show improvement over  other uncertainty quantities and methods  without using pseudo-labeling.


Strengths:
+ The quantity appears to be new and effective;
+ Validation based on error ratio and information gain is performed; comparison with other uncertainty measurements such as confidence sore and area under the margin is also  performed;
+ Algorithm limitation is discussed.

Weaknesses:
-	Some notations are not clearly explained;
-	The case study appears to be just a part of experiments, but this part is organized as a separate subsection in the methodology part. The organization may be made more  coherent;
-	Only experimental results were performed without any theoretical insight.

Significance: The topic is important and has significance in real applications.

**Questions:**

On Line 737, it mentions about the method’s “achieving an average improvement of 3.11% (the average performance gap between”. However, on most  datasets and over different L/C ratio, the improvement appears to be marginal. This average improvement of 3.11% appears a little suspicious. Is it absolute improvement or relative one?



Lines 227 and 228: What are the differences between "heterogeneity" and "diversity"?
They appear to be cyclic reasoning here.


Line 318: not very clear. there is no $i$ defined.

Figure 4: what does the bi-tuples mean for the x-axis?

On line 548:  The paper states that “… consistently increases as we progress through successive groups", but it does not appear to be the case for Pubmed and AmazonCS datasets.

**Reviewer Confidence:**

3: The reviewer is confident but not certain that the evaluation is correct

**Scope:**

4: The work is relevant to the Web and to the track, and is of broad interest to the community

---

### Decision · Program_Chairs · 2024-01-22

**Decision:**

Accept

**Comment:**

This paper proposes a memory disagreement (MoDis) technique to improve pseudo-labeling. The key idea is to use trajectory information (as entropy) as a measure of prediction uncertainty.

 The paper is generally clear, with a clean, simple idea that is well explored. The novelty isn't extraordinary, but the results are consistently (if only moderately) better than SOTA. The additional discussion, e.g. on boundary points, is nice. Researchers working on pseudo-labeling with GNNs will definitely be interested in this; it may have broader appeal to anyone working on pseudo-labeling as well.

 Strengths:
 * Consistent, though modest, improvements over SOTA.
 * Analysis showing MoDis indeed finds points near boundaries.

 Weaknesses:
 * Several reviewers expressed concerns over novelty. This technique is very similar to other previously studied ideas.
 * Utilizing trajectory information requires storing additional information, which isn't always available in practice.